# Climate change, livelihoods, gender and violence in Rukiga, Uganda: Intersections and pathways

Richard Muhumuza[1]*, Manuela Colombini[2], Pandora Zilstorf[2], Isla Collee[2], Gift Namanya[2], Joseph Katongole[1], Susannah Mayhew[2]

1 Department of Social Aspects of Health, Medical Research Council/Uganda Virus Research Institute and London School of Hygiene and Tropical Medicine Uganda Research Unit, Entebbe, Uganda,
2 Department of Global Health and Development, London School of Hygiene and Tropical Medicine, London, United Kingdom

* richard.muhumuza@mrcuganda.org

## Abstract

Climate change disproportionately affects poorer countries like Uganda, intensifying poverty and livelihood stress, which can escalate gender-based violence (GBV). Although the parent study was not designed to focus on GBV, GBV emerged repeatedly during interviews and focus groups; this paper presents a GBV-focused thematic analysis of those narratives. Particularly, we examine how GBV interconnects with poverty, shifting gender roles, alcoholism, environmental stress, and family planning dynamics. Between April and July 2021, we conducted an exploratory qualitative research study that comprised 28 focus group discussions (FGDs), comprising six-eight participants each, stratified by sex and age (18–25, 25–49, and mixed 50 + groups). Additionally, 40 key informant interviews (KIIs) were held in Rukiga district, Uganda. Purposive sampling was applied. Data were organised in NVivo 12 and analysed thematically. Participants perceived GBV, including intimate partner violence, non-partner sexual violence, child abuse, and early marriage, as widespread and normalised. Two main interconnected driver clusters emerged. First, poverty, male alcohol use, and shifting gender norms contributed to household instability. As men abandoned provider roles, women assumed more responsibilities, provoking conflict and sometimes violence from disempowered male partners. Second, environmental degradation and climate-related stressors (droughts, floods, soil erosion) worsened economic hardship, tensions, and GBV. Population growth and limited land access further strained livelihoods. While family planning was generally supported, male opposition sometimes triggered conflict. Climate change impacts are gendered, with GBV pathways shaped by shifting gender roles, norms, and relations destabilised by environmental and livelihood pressures. Addressing GBV in climate-affected communities requires gender-transformative environmental and livelihood programmes. This should include strengthened social and structural resilience to challenge inequitable gender norms and power imbalances.

which permits unrestricted use, distribution, and reproduction in any medium, provided the original author and source are credited.

**Data availability statement:** We have added an alternative email to the data availability statement: We have attached the interview guides to reflect what was asked in the field. We also attach our codebook showing the themes identified. Full original transcripts cannot be made available for further research because participants did not grant approval for this. Anonymised, excerpted data will be held in LSHTM's data repository https://datacompass. lshtm.ac.uk/ and can be requested through this site by legitimate researchers. Requests will be considered on a case-by-case basis with criteria for access including possession of ethics approval, data use agreement, purpose etc. Response times are usually 2-3 weeks. All data requests for select excerpts to clarify any ambiguities should be sent to the senior author Susannah Mayhew (Susannah.Mayhew@lshtm. ac.uk), or contact research data management at researchdatamanagement@lshtm.ac.uk.

**Funding:** This work was supported by the Robert Luff Foundation, which part-funded the LSHTM to conduct this study, and Lauren Hall and David Hearth provided a further grant to SM. The funders had no role in study design, data collection and analysis, decision to publish, or preparation of the manuscript.

**Competing interests:** The authors have declared that no competing interests exist.

## Introduction

Climate change is a global phenomenon, though its impacts are not equitably distributed [1] with the lowest income countries of the world, including Uganda, experiencing the most adverse effects [2,3]. In Uganda, the negative impacts of climate change are projected to increase [4], causing changing weather patterns and increased occurrence and severity of natural disasters, such as droughts, floods and landslides [5–7]. These events lead to widespread land degradation and damage to infrastructure, settlements and the agricultural sector, causing substantial economic and livelihood loss [5,8].

The effects of climate change and environmental degradation, and the subsequent vulnerability of people, cannot be separated from the indirect and socially mediated effects, which intersect with wider social, cultural and economic factors [9,10]. Indirect effects include the exacerbation of poverty and livelihood stresses, which in turn can trigger gender-based violence (GBV) [11,12]. Gender is among the most universal and important determinants of vulnerability to the effects of climate change [12–15]. Owing to inequities in the distribution of power and access to social and material assets, such as food, water, land, education, health, housing, employment and social capital, women and girls are more vulnerable to ecological and financial disruptions caused by climate variability [16,17]. Climate change stressors exacerbate pre-existing gender inequalities that expose women to greater risk of poverty, marginalisation and, in turn, GBV [15].

GBV is a major public health and human rights issue globally, affecting one in three women [18]. In Uganda, 44% of women aged 15–49 have experienced physical violence since age 15, and 17% of women have ever experienced sexual violence [19]. Prevalence estimates from cross-sectional studies conducted within Uganda show marked regional variations in lifetime IPV [20–23]. For instance, the prevalence of lifetime physical violence is highest in the Northern (49.2%) and Western (41.5%) regions compared to the Central (31.2%) and Eastern (41.4%) regions of Uganda [21]. The most vulnerable women are those in rural areas, those with lower levels of educational attainment, and women from low-income households [24]. Efforts to reduce levels of GBV are evident in national policies, such as the 2016 National Policy on Elimination of Gender Based Violence in Uganda, and commitment to the 2021–2025 National Action Plan for Women, Peace and Security. However, action in the policy arena is undermined by high levels of social acceptability of GBV. According to the 2022 Uganda Demographic Health Survey (DHS), almost a third (33%) of women aged 15–49 agree that a husband is justified in beating his wife for reasons such as burning food, refusing sexual intercourse, or failing to care for children [19]. GBV leads to multiple short and long-term adverse consequences for women and their children. In particular, GBV can affect physical and mental health and social well-being [25–27]. GBV is also a major contributor to reproductive health issues, including unwanted pregnancies and sexually transmitted infections [28,29].

The intersectionality of climate change with gender roles, empowerment, and GBV presents complex challenges for development practitioners and policymakers [25,30,31]. The existing literature reveals the links between climate change,

environmental degradation, and GBV to be complex and highly context-dependent. Previous studies have demonstrated how drought increases the risk for GBV throughout sub-Saharan Africa (SSA), how food insecurity is associated with intimate partner violence in central and western Uganda, and similarly, how disasters triggered by climate change are associated with GBV [32,33]. Several factors are thought to play a key role in mediating these relationships, including poverty, food insecurity, water scarcity, loss of livelihoods, male unemployment, and alcohol abuse [14,34]. Alcohol use (often triggered by poverty-related stress) is known to be a consistent risk factor for GBV, and harmful use increases the risk of violence and injury to women and children [35]. Environmental stressors may also exacerbate pre-existing risk factors for GBV, such as socioeconomic inequality, gender discrimination, rigid sociocultural gender norms, and power imbalances at various societal levels [14]. Gender norms, roles, identities and relations are recognised as important mediators between environmental stress and violence, as they are often destabilised during times of crisis [36]. For example, threatened livelihoods and increased financial and food insecurity may increase the inability of men and boys to uphold the socially ascribed role of 'provider', increasing their likelihood of perpetrating violence against women and girls [37–40].

While the nexus between gender inequality and climate change is increasingly acknowledged and becoming a focus of international policy and practice frameworks, and research [41] Most attention has been on the impact of climate disasters and natural hazards, such as floods, droughts, and earthquakes [14]. The impact of chronic slow-onset climate change, like that being experienced in Rukiga, is neglected, and the indirect pathways by which climate-related factors lead to GBV in these settings remain understudied [42].

This analysis draws on an exploratory, qualitative study that explored community perceptions of climate- and livelihood-related stressors and changing social dynamics in Rukiga District. Although GBV was not an original focus and was not explicitly probed, it was raised spontaneously and repeatedly by participants. The purpose of this study is, therefore, to examine pathways linking climate- and livelihood-related stressors, changing gender dynamics, and risks of GBV in the Rukiga District of Western Uganda.

### Theoretical orientation

This study is informed by a gendered political ecology perspective, which conceptualises climate change not only as an environmental process but as one that is socially mediated through power relations, livelihoods, and gender norms [43]. From this perspective, environmental stressors interact with existing gender inequalities and economic precarity to reshape household roles, authority, and conflict, with implications for gender-based violence [44]. We also draw on intersectional and relational gender lenses to understand how these processes are experienced differently across social positions and how threatened masculine identities may mediate pathways to violence [45,46].

## Materials and methods

### Study setting, design and parent study

Rukiga District in Western Uganda experiences slow-onset, chronic climate and environmental change. It features steep upland slopes and low-lying wetland areas, which serve as the habitat for Uganda's national bird, the Endangered Grey Crowned Crane. Most residents in Rukiga District engage in small-scale farming, traditionally on the steep valley sides and increasingly in wetland valley bottoms. Rising human activity is exerting greater pressure on local ecosystems, resulting in degraded wetlands and declining soil quality in upland farms. Additionally, access to health and especially family planning services remains limited despite high demand for these services. The regional reported lifetime prevalence of GBV is 41.5% (no data exist for Rukiga) [21]. This ecologically fragile, but environmentally important location in which human populations have poor healthcare and struggle to co-exist in a sustainable way with local ecosystems, were the reasons why Rukiga was selected by the implementing partners (working on conservation/environment, livelihoods and health).

The parent study involved formative research (reported in this paper) to identify the key health and livelihood challenges that people faced and inform intervention design. Cross-sector interventions (covering health, climate-smart agriculture and environmental protection) were then designed and evaluated (reported elsewhere) [39,47,48]. This paper reports on the findings on GBV arising from the formative study, which was designed as anexploratory qualitative research study in four parish communities within Rukiga District (Nyakagabagaba, Kitojo, Kihanga-Sindi, and Burime).

## Data collection

From 20 April 2021- 02 July 2021 focus group discussions (FGDs) were conducted in each community (28 in total) each consisting of six to eight participants, stratified by both sex (male and female) and age (18–25, 25–49, and 50 + years), except for the one of 50 + years which was mixed with an equal number of males and females. After the FGDs, 40 Key Informant Interviews (KIIs) (10 per community) were conducted. Participants were purposively selected (excluding those who had participated in the FGDs) to represent different age groups and sexes from the community with the help of community leaders. These community leaders were well known members of the community and had worked closely with the implementing partners. They were requested to identify equal numbers of men and women, from a range of backgrounds, ages and residences in the community. They were not present during the interviews, so that confidentiality could be ensured. Written and informed consent was obtained from the participants before they participated in the study. The FGDs and KIIs were conducted within the study villages in an environment that was private for both the participants and the researcher, including schools and church buildings. FGD members were chosen from the community using community mobilisers to ensure nearly all villages were represented. In each community, we had two groups in each age category, except for the 50 + group, which had only one. For the KIIs, we conducted 10 per integrated site and were mobilised in the same way as the FGDs. Data saturation was achieved quite quickly, both within and between communities, but we aimed to conduct FGDs at every site, resulting in more than necessary, and the same applied to the KIIs.

Experienced qualitative researchers (one male, RM, and one female GN) conducted the FGDs and KIIs, each lasting between 60 and 120 minutes. The study strengthened internal validity and rigour by using standardised, pilot-tested FGD and KII guides. All research assistants were well-trained. A semi-structured guide provided in our S1 Text was used to gather information on the problems affecting the community and the possible connections between these problems. It did *not* include specific questions or probes on GBV since the project was not originally focused on this, as the parent study was not originally focused on GBV; however, GBV emerged as a recurring theme during analysis.

## Data analysis

FGDs and KIIs were conducted in Runyankore/Rukiga, audio-recorded, transcribed verbatim, and translated into English. The transcripts were cross-checked by the research team against the audio recordings to ensure accuracy. GBV was not the focus of the study, which is described elsewhere [39,47], but because mention of it was ubiquitous across the sites, we undertook a thematic analysis of all excerpts where participants discussed GBV and the factors they linked to it (e.g., livelihood stress, poverty, alcohol use, shifting gender roles, and family planning-related conflict). We focused on the perceived pathways through which these stressors were thought to increase household conflict and violence. Originally, the FGDs were intended to capture community norms and responses to health and livelihood challenges. In contrast, the KIIs were intended to triangulate these community norms (from a "key informant" perspective) and to capture information on the nature of any existing collective and institutional responses to the challenges identified. In the initial identification of GBV as an emerging issue, it was clear that there was little difference in the way that FGD and KII participants spoke about GBV; therefore, for the GBV-specific analysis reported in this paper, the two datasets were analysed together using the same code frame.

Transcripts were coded and organised using NVivo 12 and analysed thematically, drawing on the research questions as well as emergent themes [49,50]. The analysis was an iterative process of discussion and revision between the wider

research team. Three researchers generated a list of recurrent codes by independently reviewing transcripts several times and making notes of key ideas and codes. After completing the initial round of coding, the researchers scrutinised the new codes and comparisons were made to check and ensure consistency. Discrepancies in the coding were re-examined, and an initial coding framework and codebook were developed and used after consensus on the final codes provided in S1 Data. While coding was primarily inductive and deductive, interpretation of the resulting themes was informed by a gendered political ecology perspective, which sensitised the analysis to participants' accounts of how environmental change intersected with livelihoods, gender norms, and power relations at household and community levels. An intersectional lens further guided the interpretation of how these dynamics were experienced differently by women and men across age groups and socioeconomic positions. We present the overall findings from the FGDs and KIIs organised by the final themes.

### Positionality and reflexivity statement

This study was conducted by a multidisciplinary team of experienced Ugandan and British researchers. The Ugandan team, who collected the data (RM, GN and JK), brought insider knowledge of the local culture, language, and community dynamics, which facilitated rapport-building, trust, and access to community members in the study sites. With no existing relationship to the community, the researchers acknowledged their outsider stance and engaged with openness. The British team members (MC, PZ, IC and SM), who were not involved in data collection, provided outsider perspectives that enhanced reflexivity during analysis and helped identify assumptions and biases that may have been embedded in interpretation. Reflexive practices, including weekly peer debriefings and internal checking of interpretations between researchers and programme staff, were used to examine how researchers' positionalities and prior experiences shaped data collection and analysis. The study did not conduct participant member checking in the conventional sense. Instead, credibility was supported through these internal reflexive processes and through dissemination meetings with community members and district stakeholders. It is during these meetings that preliminary interpretations were shared, and feedback was gathered prior to finalising the results. This combination of insider and outsider perspectives strengthened the credibility and depth of the analysis.

### Ethical considerations

This study involved human participants and was approved by The London School of Hygiene and Tropical Medicine Research and Ethics Committee (24031), Makerere University School of Social Sciences Research and Ethics Committee (MAKSS REC 10.20.447/CR) and Uganda National Council for Science and Technology (HS1137ES). We obtained informed written consent from all literate participants. For participants who were not literate, a witness confirmed that they had understood the purpose of the study. The witness was somebody who was chosen and trusted by the participant.

The project also had an ethical obligation, once GBV was identified, to support those who reported it. Based on the findings reported in this paper, the MPT developed a GBV screening tool that was used at the clinic to assess disclosure by participants. Participants who disclosed any form of violence were referred to an NGO operating in the area that offers help and support to GBV survivors (including further counselling and other support services).

## Results

### Demographic characteristics

A total of 28 focus group discussions (FGDs), each with approximately 6–8 participants, and 40 key informant interviews (KIIs) were conducted. Demographic information, however, was not consistently collected for all FGD participants. The characteristics presented here therefore reflect the 120 participants for whom complete demographic data were available. Among these, the majority were female (64.2%). Nearly half were aged 18–35 years (47.5%), with the remainder

distributed across 36–50 years (28.3%), 51–65 years (17.5%), and over 65 years (6.7%). Participants were predominantly Bukiga (96.7%). Most had primary (40.8%) or secondary education (40.0%), while 14.2% had tertiary education and 5.0% reported no formal education. Over half were married (55.8%), and farming or peasant work was the main occupation (61.7%). The majority identified as Protestant (55.8%) or Catholic (34.2%).

We first briefly describe how openly respondents spoke about GBV during the research. We then present respondents' perceptions of the gendered and livelihood-mediated pathways to GBV. These emerged in two main structural and relational processes. First, environmental degradation and stressors interplay with livelihoods and family planning use, mediated through poverty. Second, a complex interplay between lack of income (poverty), changing gender roles (with women becoming breadwinners) and alcoholism.

### Ubiquitous awareness of gender-based violence

Despite the original project was not set up to research GBV, GBV was spontaneously and consistently mentioned in almost every FGD and KII, emphasising its pervasiveness. Forms of GBV mentioned included non-partner sexual violence such as rape, intimate partner violence (physical, sexual, and emotional), violence towards children, and early marriage. Participants also discussed unintended pregnancy in relation to GBV-related dynamics, particularly conflict and coercion around family planning and reproductive decision-making (e.g., partners preventing contraception use or women using contraception in secret due to fear of retaliation). One key informant mentioned femicide. Despite a small number of men describing women being abusive towards husbands or children, most participants (both men and women) recognised that men were the primary perpetrators of physical harm towards women and children, respectively.

While we did not explore coping strategies, help-seeking, or prevention systematically, a small number of participants did describe how women try to manage violence in everyday life, particularly in the context of male alcohol use.

*"When he comes home drunk, and he talks, then you keep silent, he beats you …. So, alcoholism also causes problems." KII Female, 26 years Village B*

Throughout our data, GBV was widely acknowledged as a pervasive individual, family, and community issue that was normalised in people's lives. Although GBV was widely regarded as a norm, it was also widely acknowledged as a "problem" that negatively affects people, particularly women and girls. This is underscored by the euphemisms for GBV that many participants used, such as "no peace", "quarrels", or "conflict" throughout their experiences and perceptions. As these terms were sometimes used to describe a range of household tensions, we refer to these as conflict unless participants explicitly described violence (e.g., beating, forced sex) or clearly framed the interaction as GBV.

GBV was raised during the discussion of a wide range of factors, which are discussed below in terms of the two main emerging clusters of interrelated pathways.

### Climate-related environmental and livelihood stressors

In every village, participants spoke of the devastating effects that environmental and climatic stressors were having on their livelihoods, health, and well-being. Challenges included drought, flooding, landslides, soil erosion, reduced soil fertility, diseases affecting crops and livestock, and unpredictable seasonal and rainfall patterns. Across sites, participants perceived these environmental shocks as increasing household stress, conflict, and 'no peace', and explicitly linked this to a higher risk of violence against women and children when food and income were scarce.

*"The climate normally disturbs this village. Floods normally give us a hard time. Whenever it rains, because of the hills, all the soil is eroded, and all the crops get destroyed." KII Male, 37 years Village B*

*"Climate change affects us. There are times when it shines a lot. This causes problems in the season because crops will be destroyed by a lot of sunshine. Other times, you find that when it rains heavily, crops also get destroyed." KII Female, 48 years Village A*

Respondents spoke of how these climate-related challenges affected agricultural yields, which meant there was reduced food and income. Participants how climate-related disruptions to livelihoods (e.g., crop loss and reduced income) increased household stress and instability.

*"On top of famine, even the soils change. They lose fertility because the top fertile soils are washed away. […] When you cultivate and fail to yield well, you cannot have income. […] It causes us poverty and unrest in families because, in the absence of money, you cannot have peace in a home." KII Female, 48 years Village A*

While these pressures were often expressed as 'no peace' or 'quarrels' in the home, some accounts described this conflict escalating into physical violence against women. For example, a participant in one FGD of younger women clearly articulated the connections she saw between the landslides, which destroyed her subsistence farm, leading to a lack of food, which caused her husband to beat her in anger. For her, the "chaos" was the direct result of her farmland being destroyed by landslides:

*"When the environment gets worse, say, if landslides destroy my garden, that means that even if I cultivate there, I will not get what to eat. Then, the moment I fail to get what to cook for the family, now, even if the man is away and comes back and finds that I have not cooked, he will say, my children are starving, anger makes him beat the woman, the woman transfer the anger to children and also beat them, now you find the whole family chaotic not because you lacked what to cook but because the land was destroyed by landslides." FGD Females, 18-25 years Village C*

Another frequently mentioned factor that affected the loss of productive farmland and put further pressure on livelihoods was population increase, as large families struggled to produce sufficient food to feed themselves.

*"People in ancient days […] had plenty of land and would cultivate, let their soils rest, and regain fertility, but now, maybe because of the overproduction of children, the land is scarce. So, I see this as a great problem, and it is getting worse." FGD Males, 26–50 years Village A*

This pressure further exacerbated the livelihood stresses caused by climate change in two ways: by multiplying the number of people dependent on the same tract of land, and by leading to environmentally unsustainable farming practices.

*"On the environment, people cut down all the trees on the hills, leaving the land bare. When it rains, water from the hills combines with that from the houses, and it floods the valleys, destroying the crops. When a person has nothing to eat, that's when they go and burn the wetland to catch mudfish for food." KII-Male-48 years-Village D*

*"Even if a person tries to dig and grow crops, due to many children to feed, he cannot get any food balance to take to the market for sale. They eat it all. Because they do not have an alternative, they end up encroaching on some reserved lands and cultivating them."* KII-Male-37 years-Village B

*"Because of limited land and growing crops on the same land, the soil loses fertility. So even when you plant your crops, they don't yield, and when the sunshine comes, they all dry up and decay in the rainy season … What causes*

*that is poor farming, wetlands were seriously cultivated, and when there is water runoff, it affects everywhere. People don't mind planting trees that reduce the amount of water.*" FGD-Females-26–50 years-Village A

One long-term solution that many respondents identified is better access to family planning (FP):

"*[…] you should make sure you get us family planning methods that fit our health so that our families are put in good condition.*" FGD Females 26-50 years Village B

Although disagreements surrounding women's use of family planning methods were raised by many participants as a specific source of conflict between women and their husbands, most respondents (men and women) expressed positive attitudes towards FP. Where men did not support FP, the reasons for their rejection included a desire to have more children and a lack of understanding of FP methods. One woman explained that:

"*People understand differently. Some like and support it [FP] while others do not support it. Still, some women do family planning secretly without the knowledge of their husbands.*" FGD M/F, 51-70 years Village D

Whilst some women overcame opposition from their husbands by accessing FP services in secret, the *possibility* of conflicts over FP usage acted as a deterrent to other women.

"*Most women do not tell their husbands about family planning. They just wake at once and go for family planning without telling their husbands... as the man discovers it, they both resort to quarrels. Now, as their colleagues discover this, they also fear joining family planning.*" FGD M/F, 51–70 years Village D

Overall, participants framed environmental change as worsening livelihood insecurity, which then fuelled household conflict and violence.

## Poverty, alcohol use, and changing gender roles

When describing experiences and perceptions of poverty and its perceived consequences including poor diets, sickness, and lack of ability to pay for healthcare and transport to medical facilities, most respondents explicitly identified poverty as a direct cause of GBV. In this cluster, participants perceived poverty and men's harmful alcohol use as direct triggers for partner violence, with conflict often intensifying when women took on breadwinning roles, and men felt their authority and provider role was undermined.

"…*when poverty knocks on your door, you start quarrelling and fighting, and consequently, the family ends like that.*" KII Female, 53 years Village C

"*When a child gets sick, you do not have money. When you fail to get food, you fight with the woman. So, I think failure to get daily income is the leading cause of our problems.*" FGD Males, 26–50 years Village B

Poverty was also identified as a trigger for alcohol use: "*You find that the man has no income. He has resorted to spending all his time at the bar*" (FGD Female, 18–25 years Village C). Harmful alcohol consumption was in turn, perceived as a direct trigger for GBV:

"*And men who drink alcohol, in their homes, there is a lot of quarrels.*" KII Female, 49 years Village A

A common narrative emerging amongst participants was that harmful alcohol use by men directly increased household financial and economic instability. Through spending the family's already limited income on alcohol, men redirected

income away from food, basic household goods, and children's school fees. Some respondents described how men steal their wives' harvests or income to purchase alcohol. This was cited as a major source of conflict between spouses:

*"The money that is intended to support the family is often found to have been spent at the bar, leading to conflicts within the family." FGD Males, 18-25 years Village C*

Poverty was not only identified as a consequence of harmful alcohol consumption but also a driver, largely mediated through unemployment and feelings of idleness. In this way, respondents painted a picture of a vicious cycle between poverty, alcohol dependency, and GBV:

*"Men don't have [anything] to do, so for every coin they get, they direct it to the bar. If they had jobs and they got the money, they would be planning for their families. So, poverty is one cause of alcoholism."* KII Male, 67 years Village D

*"Development in the homes of men who drink a lot of alcohol is very low […] those who stay in the bar, cannot do anything in their homes" KII Female, 49 years Village A*

Alcohol use and GBV were also discussed in relation to changing gender roles and norms. Participants gave clear accounts of patriarchal gender roles, where men were seen as the respected decision-makers. By contrast, the roles and expectations for women included acting as homemakers and being responsible for the well-being of the children and family.

*"The men are the ones who are supposed to make decisions because they have more respect than the women.*" FGD Female, 18–25 years Village D

*"We know men are heads of families, but we women have a lot of activities in the home." FGD Females, 26–50 years Village B*

Yet, despite these traditional characterisations of gender roles and expectations, respondents described a changing reality in which men were accused (by both men and women) of abandoning their duties as family heads, resulting in the responsibility for income and accompanying decision-making shifting to the shoulders of the women:

*"Under normal circumstances, the head of the family is the man. But these days, you find a man does not want to know about the well-being of his family. The woman wakes up early and prepares for the children to go to school, and the man does not want to know. It is the woman that takes care of everything." FGD Females, 18-25 years Village A*

To provide an income to support their families in the absence of male breadwinning, many participants described the extra efforts of women to provide for their families, often characterising alcoholism as a reason for women taking over abandoned responsibilities.

*"A man wakes up early and goes to the bar, fails to go to cultivate [the farm] with his wife, and drinks the whole day. Isn't [it] here that senses disappear? So, that is why a woman leads such a family because such a man is senseless." FGD Males 51–75 years Village B*

*"Even in our families, change came because when a woman finds out that her husband is an alcoholic and not responsible, she takes up the responsibility of the family, digs for money, and pays school fees for her children, can't wait for the man to do it. So, change came and now women woke up." KII Female, 48 years Village A*

Meanwhile, the increasing engagement of women in income generation for their families threatens the traditional power balance, with participants describing the male partners of such women as lacking authority.

*"At times, you find that you have a wife who has a business and has more money than you. So, such a woman will rule you because, for you, you stay in the village and have nothing to do." FGD Males, 18–25 years Village B*

*"…the man is only thinking about alcohol, when the child asks for a book, the father tells the child to ask the mother because the mother is the one who works, and therefore, she has some money. So, the man does not have the authority to issue some directives in his home just because he does not have money."* FGD Female, 18–25 years Village A.

However, women stepping into the role of breadwinner does not earn her the respect that is traditionally given to men in such positions. Rather, she is regarded with suspicion because the switching of roles is uncomfortable, undermining the position of men as the quotes above indicate phrases such as "such a woman will rule you" and "the man does not have the authority to issue some directives in his home". This suspicion was reported to result in serious, violent consequences for some women, as though men could not accept that women could legitimately earn money through hard work and efficiency but must instead have been unfaithful – the ultimate humiliation for a husband:

*"You find that the man has no income. He has resorted to spending all his time at the bar. Now, for me, when I cultivate a given garden, I keep saving some money so that when it accumulates, I can also buy some land for cultivation. When the man discovers that you have bought a piece of land, he can nearly beat you to death. He asks you where you got the money from. Did you adulterate for it? Now, you find instabilities rising." FGD Female, 18-25 years Village C*

Yet the impact of such violence would not only be felt by the women, but the whole family, as this male participant explained:

*"It does not only disturb the woman, but also the income of a home dies completely when domestic violence happens." KII* Male, 45 years Village C).

These extracts indicate how alcoholism can be fuelled by poverty, triggering GBV, but also how alcohol abuse is connected with changing gender roles. Alcoholism was seen as both as a cause and a consequence of perceived male disempowerment and a failure to meet traditional markers of masculinity, including the perceived failure to provide financially within household settings. Consequently, this led to women taking leading roles as decision-makers and breadwinners for their families, but often with violent consequences. Participants therefore described a reinforcing cycle in which poverty and alcohol use increased conflict and violence, while shifting gender roles shaped when and how that violence occurred.

## Discussion

Interpreted through a gendered political ecology lens, these findings illustrate how slow-onset climate change operates through livelihoods and social relations to reconfigure gender roles and power within households, producing conditions under which GBV is more likely. This study examined locally held perceptions of the relationship between climate- and livelihood-related stressors and changing gender dynamics, including the risks of GBV in the Rukiga District of Southwestern Uganda. Our findings reveal a complex interplay of factors, including climate-related livelihood stressors, poverty, harmful alcohol use, change in gender roles and norms and gender-based violence. The perceptions of the interconnections, as reported by our respondents, are summarised in Fig 1.

Poverty was seen as a pervasive direct trigger for GBV, and climate change exacerbates poverty through environmental shocks and chronic change that leads to food insecurity and livelihood stressors. Growing populations, partly because

PLOS Global Public Health

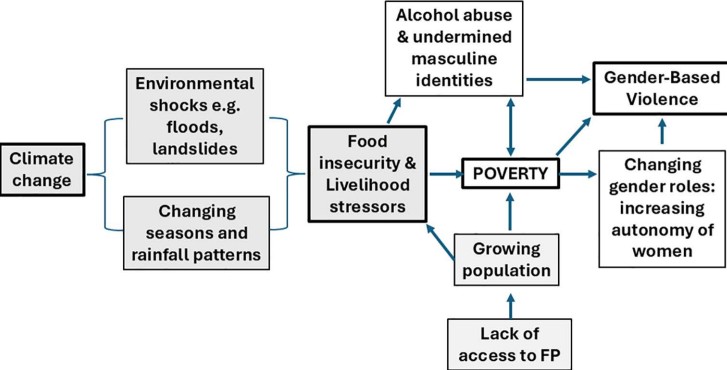

**Fig 1. Perceived connections and pathways between climate change, livelihoods and gender.**

of a lack of access to FP services, also contribute to food shortages and poverty. Poverty was seen as a trigger for harmful alcohol use, which was also identified as a direct cause of GBV, and particularly of partner violence by men against their wives. In the context of rising poverty and increased male alcohol abuse, many women were taking on income generation roles themselves in order to feed, clothe and educate their families. These changing gender roles in relation to women's economic independence were seen to result in a concomitant reduction of male decision-making roles. This also led to family tensions that were sometimes perceived as resulting in GBV. We discuss these interlinkages in relation to wider literature.

## Interlinkages between climate change, environmental and livelihood stressors, and poverty

The findings of our research demonstrate how climate change, extreme environmental fragility and economic and livelihood insecurity are inextricably linked. When combined, they exacerbate one another to increase poverty. Poverty emerged as the most prominent direct trigger for GBV by creating acute and chronic household and community stress, thereby indirectly linking climate change to GBV. These findings support the work of others, notably Carney et al. (2020), who show how the effects of both chronic climate change (prolonged dry seasons) and climate induced environmental disasters in Uganda led to the destruction of natural resources that underpin livelihoods, driving resource scarcity and poverty, thereby leading to increased risk of GBV [39,51]. A systematic review of natural hazards and disasters and their link with violence against women and girls [14,47] identified important pathways which, though not specific to chronic or slow-onset climate change, do mirror our findings. In particular, they find that extreme climate events increase economic stressors that lead to violence and exacerbate existing drivers of GBV. Other work in Uganda, not specific to climate, also identified poverty and economic insecurity [52] as well as alcohol abuse as significant drivers of GBV [53]. Livelihood pressures and alcohol abuse are well-known triggers of violence, evidenced in wider literature across multiple settings, including sub-Saharan Africa [40,54].

## Alcohol abuse and masculine identities

Many respondents identified men's alcohol use as a key factor mediating the relationship between environmental degradation, livelihoods and violence. This is supported by Uganda's most recent DHS data, which found that women whose husbands are often drunk are much more likely (84%) to experience GBV than women whose partners do not drink (45%) [55]. Whilst alcohol itself is not a "necessary or sufficient cause" of GBV [56,57], it is recognised that alcohol use, especially excessive consumption, influences the frequency and severity of GBV [58,59]. However, the mechanisms underlying this association are still contested. In this study, participants recognised alcohol abuse by men to be both a symptom and

PLOS Global Public Health

cause of financial stress, unemployment, and feelings of inadequacy following a failure to fulfil their gendered role of provider. Adopting a gendered perspective illuminates how alcohol misuse may intersect with gender norms to elicit violence, such as in contexts where drinking is associated with harmful notions of masculinity [57], or used as a coping mechanism. In these contexts, alcohol use may be used to assert harmful masculine identities, especially in the absence of other traditional denotations of masculinity, such as employment [57].

**Changing gender roles: Women's economic participation**

The participants in our study relied heavily on agricultural practices and natural resources for their livelihoods, and the disruptive impact of climate change on these sectors led to destructive social and economic systems. Participants illustrated how environmental stressors, including crop failure and loss of livelihoods, were forcing women to adopt new roles and responsibilities out of necessity. As men turned to alcohol in frustration, women increasingly acted as primary income earners, with increased decision-making power over household resources. Participants also described how these gender role reversals were a source of household conflict and led to partner violence. This is consistent with other literature, which shows that when the social fabric and corresponding gender roles within communities are dramatically altered by the deterioration of local economies, dispossession of land, loss of local livelihoods and degradation of natural resources, this can give rise to GBV (Alston & Whittenbury, 2012; Weltbank, 2012; Namasaka, 2014; Barcia, 2017b). Carney et al. (2020) highlight how livelihood disruption can cause social vulnerability, community wide livelihood and economic insecurity and acute and chronic stress, which strengthens the conditions for gender-based violence to occur [51]. Previous research also suggests that GBV is common in transitional societies undergoing sociocultural shifts away from traditional patriarchal values towards gender equality. This has been studied in the Ugandan context, where domestic violence in Wakiso District was found, in part, to be a response to unequal balances of power in a shifting socio-economic environment [60] and appears to be what we are seeing in Rukiga as well.

Our findings contradict other literature, however, that suggests natural disasters may induce positive shifts in gender relations through enabling increased female economic participation in post-disaster settings [61], which some authors have seen as acting as potential 'windows of opportunity' to drive social change [36,62]. Whilst women's economic empowerment interventions, such as microfinance programs and cash transfers, have been suggested to lower women's GBV risk [63,64], evidence for their effectiveness is varied. Evidence from the MAISHA study for prevention of intimate partner violence (IPV) in Tanzania found that whilst an overall increase in women's income was protective against IPV, those who financially contributed more than their husbands were at greater risk of sexual and physical violence [65]. These effects may be explained with reference to gender theory. *Gender role strain theory* hypothesises that men who perceive themselves as failing to fulfil expected male duties, such as providing for their families, experience poor psychological outcomes, which may increase the infliction of violence against their partners [66]. Similarly, *gendered resource theory* posits that in traditional households where women act as breadwinners, husbands use violence to compensate for feelings of disempowerment [67]. Both may be applied here to explain how increased economic participation by women in Rukiga challenges hegemonic masculine identities.

Despite the challenges of GBV faced by women like those in Rukiga, our findings show that they possessed considerable capacity to adapt to environmental degradation and mitigate effects on livelihoods. Adaptive strategies outlined by participants included adopting the dual burden of caretaker and primary income earner, engaging in alternative livelihoods, and joining savings groups. These findings support a growing body of research that identifies women as key agents of adaptation to climate change-related challenges [36]. Nevertheless, the invisible yet dangerous pathways to GBV that we identify must be carefully considered by programs seeking to empower women through encouraging economic participation, to avoid exacerbating pre-existing challenges.

## Limitations

Two main limitations must be acknowledged. Firstly, as GBV was not probed, discussions of typologies, coping, and prevention reflect what participants raised spontaneously and should not be interpreted as comprehensive. Therefore, this data provides an incomplete picture of GBV in Rukiga district. Further qualitative research specifically investigating GBV must be conducted to further understand the complex pathways between environmental degradation, climate change, economic instability and violence.

Secondly, the effects of sampling, social desirability, and recall bias must be considered in all qualitative research. Here, key informant interviews (KIIs) were conducted with individuals in positions of power within their communities. Given that experiences of gender-based violence (GBV) differ according to social status, poverty, and education, these perspectives may not represent those of less-powerful individuals, although conducting focus group discussions (FGDs) also helped to mitigate this risk. However, potential imbalances in wealth and educational power may have been evident between the interviewers and participants, particularly in the FGDs, which could have facilitated acquiescence and social desirability bias, where participants felt compelled to respond in certain ways. Again, triangulation with the KIIs assisted in mitigating this, and we noted the similarities in responses. Nevertheless, further research is necessary to adequately understand the breadth of experiences of GBV in Rukiga.

## Conclusions

As the effects of climate change continue to threaten global human and environmental health, it is increasingly important to understand how environmental changes will disproportionately impact vulnerable populations and exacerbate pre-existing vulnerabilities. Our findings revealed a strong perception of worsening household and livelihood stress linked to climate change, which seems to be reshaping gender roles and dynamics within the family and, in some cases, contributing to increased conflict within households.

This demonstrates the importance of considering the gendered impacts of climate change in vulnerable communities, building social and structural resilience to effectively prevent GBV. Promoting environmental and livelihood programmes that are gender-transformative would help to address gender norms and power inequities so that both men and women are able to support their families, despite the challenges of climate change and environmental degradation.

From a gendered political ecology perspective, interventions that focus solely on livelihoods or environmental resilience without addressing gender norms and power relations risk reproducing or exacerbating GBV. Gender-transformative climate and livelihood programmes must therefore engage both women and men, address harmful masculinities, and strengthen social as well as economic resilience.

## Supporting information

**S1 Text. Topic guide final.**
(DOCX)

**S1 Data. Coding framework.**
(XLSX)

**S1 File. COREQ checklist.**
(DOCX)

**S1 Checklist. Inclusivity-in-global-research-questionnaire.**
(DOCX)

## Acknowledgments

We acknowledge the contributions of the MRC/UVRI and LSHTM Uganda Research Unit, the Margaret Pyke Trust, Rugarama Hospital, the International Crane Foundation, and the Endangered Wildlife Trust for their support services, which made this study possible. We are particularly grateful to all the study participants, not forgetting the mobilisers, for the time and information they shared with us.

## Author contributions

**Conceptualization:** Richard Muhumuza.

**Data curation:** Richard Muhumuza, Gift Namanya, Susannah Mayhew.

**Formal analysis:** Richard Muhumuza, Pandora Zilstorf, Gift Namanya, Joseph Katongole, Susannah Mayhew.

**Funding acquisition:** Susannah Mayhew.

**Investigation:** Richard Muhumuza, Susannah Mayhew.

**Methodology:** Richard Muhumuza, Manuela Colombini, Pandora Zilstorf, Isla Collee, Gift Namanya, Joseph Katongole, Susannah Mayhew.

**Project administration:** Richard Muhumuza, Susannah Mayhew.

**Resources:** Richard Muhumuza, Susannah Mayhew.

**Software:** Richard Muhumuza, Susannah Mayhew.

**Supervision:** Richard Muhumuza, Susannah Mayhew.

**Validation:** Richard Muhumuza, Isla Collee, Gift Namanya, Joseph Katongole, Susannah Mayhew.

**Visualization:** Richard Muhumuza, Manuela Colombini, Isla Collee, Gift Namanya, Joseph Katongole, Susannah Mayhew.

**Writing – original draft:** Richard Muhumuza, Manuela Colombini, Pandora Zilstorf, Isla Collee, Gift Namanya, Joseph Katongole, Susannah Mayhew.

**Writing – review & editing:** Richard Muhumuza, Manuela Colombini, Pandora Zilstorf, Isla Collee, Gift Namanya, Joseph Katongole, Susannah Mayhew.

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
