## [Decision Letter · Decision Letter 0]

11 Dec 2025

PGPH-D-25-03150

Climate change, livelihoods, gender and violence in Rukiga, Uganda: intersections and pathways

Dear Dr. Muhumuza,

Thank you for submitting your manuscript to PLOS Global Public Health. After careful consideration, we feel that it has merit but does not fully meet PLOS Global Public Health’s publication criteria as it currently stands. Therefore, we invite you to submit a revised version of the manuscript that addresses the points raised during the review process.

We look forward to receiving your revised manuscript.

Kind regards,

Zahra Zeinali, MD MPH DrPH

Academic Editor

Journal Requirements:

Please include a complete copy of PLOS’ questionnaire on inclusivity in global research in your revised manuscript. Our policy for research in this area aims to improve transparency in the reporting of research performed outside of researchers’ own country or community. The policy applies to researchers who have travelled to a different country to conduct research, research with Indigenous populations or their lands, and research on cultural artefacts. The questionnaire can also be requested at the journal’s discretion for any other submissions, even if these conditions are not met.  Please find more information on the policy and a link to download a blank copy of the questionnaire here: https://journals.plos.org/globalpublichealth/s/best-practices-in-research-reporting. Please upload a completed version of your questionnaire as Supporting Information when you resubmit your manuscript.

Please provide separate figure files in .tif or .eps format.

In the online submission form, you indicated that “Data will not be shared publicly due to the data-sharing policy of the LSHTM requiring a prior data-sharing agreement. However, the dataset containing the data supporting the study findings in this report can be obtained upon request from the corresponding author.”All PLOS journals now require all data underlying the findings described in their manuscript to be freely available to other researchers, either1. In a public repository,2. Within the manuscript itself, or3. Uploaded as supplementary information.This policy applies to all data except where public deposition would breach compliance with the protocol approved by your research ethics board. If your data cannot be made publicly available for ethical or legal reasons (e.g., public availability would compromise patient privacy), please explain your reasons by return email and your exemption request will be escalated to the editor for approval. Your exemption request will be handled independently and will not hold up the peer review process, but will need to be resolved should your manuscript be accepted for publication. One of the Editorial team will then be in touch if there are any issues.

Additional Editor Comments (if provided):

Reviewers' comments:

Reviewer's Responses to Questions

**Comments to the Author**

1. Does this manuscript meet PLOS Global Public Health’s publication criteria? Is the manuscript technically sound, and do the data support the conclusions? The manuscript must describe methodologically and ethically rigorous research with conclusions that are appropriately drawn based on the data presented.

Reviewer #1: Partly

Reviewer #2: Partly

Reviewer #3: No

2. Has the statistical analysis been performed appropriately and rigorously?

Reviewer #1: N/A

Reviewer #2: Yes

Reviewer #3: N/A

3. Have the authors made all data underlying the findings in their manuscript fully available (please refer to the Data Availability Statement at the start of the manuscript PDF file)?

Reviewer #1: No

Reviewer #2: No

Reviewer #3: Yes

4. Is the manuscript presented in an intelligible fashion and written in standard English?

Reviewer #1: Yes

Reviewer #2: Yes

Reviewer #3: Yes

5. Review Comments to the Author

Reviewer #1: Full Title:

Manuscript full title does not match with the short title. Full title reads "Climate change, livelihoods, gender and violence in Rukiga, Uganda: intersections and pathways". While short tile reads "Climate Change and Gender Based Violence". 'gender based violence' may not necessarily mean the same as 'gender and violence'. Authors should consider revising the wording in the full time if they meant gender based violence.

Abstract:

Inconsistency in FGD size, harmonize to consistent range across the manuscript. Author said "Between April and July 2021, we conducted 28 focus group discussions (FGDs), comprising 6-8 participants each (line 29-30" and in methods author said "From 20 April 2021- 02 July 2021 five focus group discussions (FGDs) were conducted in each community (28 in total) each consisting of four to six participants (lines 135-136)".

clarify the CBV emergent theme. You said "This study, though not originally intended to focus on GBV, examines how it interconnects with poverty, shifting gender roles, alcoholism, environmental stress, and family planning dynamics." (lines 26-28). Consider adding a statement signalling GBV emerged inductively during data colletion and/or analysis.

Methods:

Revise the methods section to ensure the study can be reprodcible, and signal reliability of findings.

What study design did you use? not clear

Author said participants were " purposively selected... with the help of community leaders" (lines 140-141). Clearly elaborate the eligibility criteria and how the gatekeepers' influence was mitigated, and proper justification why 28 FGDs and 40 KIIs were sufficient. Talk about saturation, was maximum variation considered? and how?

Results:

Tag all quotes with data source (FGD or KII), sex, age to evidence diversity across the groups.

Make sure all quotes are in clear quotations marks (lines 220-222). fix that for the entire results section and be consistent.

Authors said "When describing their experiences and perceptions of poverty and its associated consequences including poor diets, sickness, and lack of ability to pay for healthcare and transport to medical facilities, most respondents explicitly identified poverty as a direct cause of GBV:" (lines 311-314). Revise the wording on participants' perceptions to avoid implying causality from qualitative data. Check the entire document for this including the abstract lines 36 to 41.

Ethics:

Include ethical committee name that gave ethical clearance for the study, also include the reference number and date.

describe safeguardings and referral procedures followed in the study if any.

Conclusion:

The concept for this paper is timely and relevant. However several important elements require revision before the manuscript can meet PLOS Global Public Health Standards. Work on the clarity and consistency of the methods (study design was not clearly mentioned, there are several qualitative designs one can use, e.g. phenomenology, case study, etc. what design did you use?). PLOS Global Public Health guidelines on data sharing require that you provide some de-identified data, nevertheless authors stated that they would share data and the justification for that leaves much to be desired.

Reviewer #2: 1. Kindly mention the methodological orientation adopted for the study?

2. Discrepancy between number of participants in FGD mentioned in abstract and methods – (6 – 8 in abstract and 4 – 6 participants in methods)…Kindly make it uniform

3. Additional context on domestic violence and related statistics can be added in study setting

4. Details on steps taken to ensure internal validity/rigor to be mentioned – member checking, reflexivity

5. Give details of the parent project briefly

6. Any conceptual model/framework adopted to guide data generation/analysis?

7. What efforts were taken to address/refer victims of GBV once disclosed?

8. Socio - demographic details of the respondents could be added for better interpretation

9. Key themes are restated multiple times; Many dimensions of GBV (more details on each typology, coping strategies, prevention, etc) not elicited

Reviewer #3: Overall Comments

The paper takes a qualitative approach to “examine locally held perceptions of the relationships between climate and livelihood-related stressors and changing dynamcis, including the risk of Rukiga district. Climate change remains a global threat, with many countries and communities within Africa, ill prepared to adapt and mitigate the consequences. The paper is an attempt to paint a picture of climate-related impacts, particularly how gender-based violence, a persistent public health, socioeconomic and development issue is shaped by and influencing social, economic and environmental stressors.

In its current form, the paper need to be strengthened to get it to be sufficiently robust for publication in PLOS Global Public Health. The paper needs to be strengthened in at least three ways:

1) Overall, the paper needs to better contextualise their goal. Authors state in line 115 to 117, that their purpose is to understand locally held perceptions of the relationship between climate and livelihood-related stressors, and in several other sections, indicate make clear that, their original intention was not GBV, but undertook a thematic analysis on the latter. This can be confusing making it difficult for readers to follow. Authors need to clarify their focus – if it is on GBV, they may consider better contextualising their paper, especially in the introduction.

As part of contextualising, authors may consider highlighting the initial primary research focus – this helps to provide context for readers to begin to appreciate how and why GBV took center-stage during the analysis. In doing so, it also provides an opportunity for authors to properly situate their contributions to the literature.

Other minor issues include:

• Authors make claims about projected exponential increase (line 51-52) and yet, do not support with any data. Similarly, authors may want to consider revising the sentence, as it appears redundant.

• In line 55-57, it argued “Uganda’s vulnerability to climate change and climate-sensitive disasters is extremely high – it is not immediately clear to readers what this means. By which benchmark or metric are authors assessing Uganda’s vulnerability. Authors may consider revising to ensure clarity (also see lines 108-110 for punctuation issues).

• Lastly, the study takes place in Rukiga District – it would be helpful if authors provided some additional background context. Will the results be different, if the study was conducted in a different district rather than Rukiga? Basically, some discussions of the rationale and/or choice of the selected district is be useful.

2) Overall, authors need to improve their methods by revising and clarifying, some of the sections. For example, under study setting (line 128-130), it is not clear if the concluding sentence is provided additional context for the prior statement. Authors may want to revise for clarity purpose.

I. Reconcile the number of participants for FGDs – in the abstract, authors indicate 6-8 people form a FDG and in line 136, it says “…each consisting of four to six participants,…”.

II. For both FGD and KII, it is useful to indicate and/or describe the demographic/characteristics of the people participating in the study (Perhaps, authors could outline their demographics by sex and age, and any other stratifier in the results section in a tabular format. How were participants selected, especially among the FGD participants?

III. On ethics statement, although the data emanates from key informants and community members, authors do not indicate whether they sought ethnical approval for their study. If ethics was obtained, it is useful to indicate so.

IV. Regarding data collection, lines 172 to 173, authors indicate that “discrepancies in the coding were re-examined…”. It useful to explain how the independent assessor resolved discrepancies and reached consensus.

V. In the data collection section (line 155 to 157), authors indicate that they “undertook a specific analysis of what participants said about GBV”. However, in the results, it is often not clear, the specific thematic issues or results arising from this analysis. Related to this and linked to the analysis, it is not clear to readers how the two main clusters (line 188 to 191) link to GBV. While lines 193 to 212, describe nature of GBV, for the most parts (for example, line 213 to 308), it is difficult to follow how GBV is an interconnector in the results being discussed. At times, it difficult to see, where the analysis departs from its original intended goal. Were the issues around climate change and environment among others emergent from the data?

3) Overall, the results section outlines some very interesting insights. However, I do feel this section can be deepened. In many instances, the narratives are often not immediately supported by the relevant quotes, linking to GBV.

• In line 230 – 323, authors reflect that the disruption to livelihoods leading to family instabilities and conflict, demonstrate how GBV is triggered. This assumption is challenging to sustain, considering that “unrest in families” and not having “peace in a home” do not necessarily connote GBV. Similar reflections are presented at line 306 (“...they both resort to quarrels…”), lines 316 to 320 (…start quarrelling and fighting…”) and (“…you fight with the woman”).

• Although authors indicate these are “euphemisms for GBV” (line 208) that participants use – without critical analysis, we risk painting a picture that may not be correct. For example, will readers be correct to assume, that in Ugandan context, such referencs always mean GBV?. To avoid readers assuming without appropriate understanding of context, authors may consider, making explicit any additional nunaces related to the quotations or contexts for this pharses, to clarify and make the links to GBV much clearer.

Minor

• Line 199 – please clarify how and why unintended pregnancies is considered a form of GBV

• Line 208 to 209 – revise sentence – it is not clear what authors mean by throughout their experiences and perceptions

• Line 211 – “GBV was raised during the discussions of a wide range of factors” – perhaps, useful to outline the contexts which GBV was raised

6. PLOS authors have the option to publish the peer review history of their article (what does this mean?). If published, this will include your full peer review and any attached files.

**Do you want your identity to be public for this peer review?** For information about this choice, including consent withdrawal, please see our Privacy Policy.

Reviewer #1: **Yes:** Bristol Moonga Ntebeka

Reviewer #2: No

Reviewer #3: No

Figure Resubmissions:

---

## [Editor Report · Decision Letter 1]

24 Feb 2026

PGPH-D-25-03150R1

Climate change, livelihoods, gender and violence in Rukiga, Uganda: intersections and pathways

Dear Dr. Muhumuza,

Thank you for submitting your manuscript to PLOS Global Public Health. After careful consideration, we feel that it has merit but does not fully meet PLOS Global Public Health’s publication criteria as it currently stands. Therefore, we invite you to submit a revised version of the manuscript that addresses the points raised during the review process.

We look forward to receiving your revised manuscript.

Kind regards,

Zahra Zeinali, MD DrPH

Academic Editor

Journal Requirements:

Additional Editor Comments (if provided):

Dear Dr. Muhumuza,

Thank you for submitting the revised manuscript. The paper shows substantial improvement, with many of the reviewers’ core concerns addressed. Notably, the framing has been strengthened through a dedicated theoretical orientation, the parent project is more clearly articulated, the Ugandan context is better developed, and euphemistic language (such as “no peace” and “quarrels”) is handled more carefully to avoid conflating conflict with gender-based violence (GBV).

Before proceeding to acceptance, we request a minor revision to address a few remaining issues. These items pertain to qualitative reporting rigor and consistency. If these concerns are addressed clearly, we anticipate moving forward without further external review.

Required revisions

1) Rigor and reflexivity

In your response to the reviewers, you emphasize pilot testing and team debriefing as strategies to enhance rigor. These are helpful, but they do not fully address the reviewer’s specific request regarding member checking and reflexivity/positionality.

Please revise the Methods (or a short reflexivity subsection) to include:

* Member checking: State explicitly whether participant/member checking was conducted.

* If yes, describe what was done and at what stage (e.g., returning summaries, validating themes, community feedback session).

* If no, state this transparently and briefly justify why (e.g., feasibility, safety/ethics, rapid design), and strengthen alternative credibility strategies (e.g., audit trail, reflexive memoing, peer debriefing, negative/discordant case consideration, triangulation logic where relevant).

* Reflexivity/positionality: Add a short paragraph describing:

* who collected the data (roles, relevant identities if appropriate),

* relationship to participants/communities (including any implementing partners),

* steps taken to mitigate power dynamics and interpretive bias, and

* how reflexive practices were incorporated (e.g., memoing, team discussions, analytic decisions).

2) Title consistency

There appears to be a remaining mismatch between the full title and the short title, and/or between the manuscript file and the submission system metadata (e.g., “gender and violence” vs “gender-based violence”). Please ensure:

* the full title and short title are aligned, and

* the title in the submission system exactly matches the title shown in the manuscript file.

3) Data Availability statement

We acknowledge the rationale for not sharing qualitative transcripts publicly, given their sensitivity and potential impact on participant safety. However, the Data Availability statement should be strengthened to maximize transparency and align with the journal's expectations.

Please revise the statement to specify clearly:

* what can be shared (e.g., de-identified excerpt set, codebook, analytic framework, thematic matrix, or other non-identifiable materials), even if full transcripts cannot be shared;

* how requests are handled (who adjudicates requests—named office/role/institution rather than an individual where possible);

* criteria for access (e.g., ethics approval, data use agreement, purpose/risks); and

* process/timeline for responding to requests.

If you cannot share any excerpted qualitative data at all, please state this explicitly and provide the strongest feasible alternative (e.g., codebook and analytic outputs).

4) Consistency checks

Please do a final consistency check for the following:

* FGD size: Ensure the stated group size is consistent throughout the manuscript (abstract, methods, results, tables/appendices if any).

* Sampling/geography wording: Ensure terms like “communities/parishes” and the number of sites are consistent throughout.

* Referral pathway wording: Ensure the safeguarding/referral description is consistent, clear, and accurately reflects what was implemented in practice (and does not imply default referrals that were not part of protocol).

5) Clean tracked-changes + clean manuscript

To facilitate production, please upload:

* a clean, revised manuscript (no tracked changes), and

* a tracked-changes version that is carefully checked for accidentally duplicated paragraphs or repeated blocks inserted during revision.

Optional but recommended refinements

* Design labeling clarity: Consider refining the study design description (e.g., “rapid qualitative inquiry” or “rapid assessment with thematic analysis”) and briefly clarifying the operational meaning of “rapid” (e.g., timeline or analytic approach) to reduce ambiguity for qualitative readers.

* Causal language sweep: Continue to avoid causal claims and ensure that the Results and Discussion sections consistently reflect perceived pathways (for example, “participants described…” or “participants perceived…”).

Reviewers' comments:

 Figure Resubmissions:

---

## [Editor Report · Decision Letter 2]

23 Mar 2026

PGPH-D-25-03150R2

Climate change, livelihoods, gender and violence in Rukiga, Uganda: intersections and pathways

Dear Dr. Muhumuza,

Thank you for submitting your manuscript to PLOS Global Public Health. After careful consideration, we feel that it has merit but does not fully meet PLOS Global Public Health’s publication criteria as it currently stands. Therefore, we invite you to submit a revised version of the manuscript that addresses the points raised during the review process.

We look forward to receiving your revised manuscript.

Kind regards,

Zahra Zeinali, MD MPH DrPH

Academic Editor

**Journal Requirements:**

**Additional Editor Comments (if provided):**

The manuscript has improved considerably over the course of revision. The revised version demonstrates meaningful attention to the reviewers’ and editor’s comments, particularly in strengthening the framing, clarifying the methodological approach, expanding reflexivity, improving the discussion of safeguarding and referral processes, and providing a more nuanced interpretation of the relationship between livelihood stress, household conflict, and gender-based violence.

However, a small number of issues remain outstanding and should be addressed before the manuscript can be accepted.

First, the issue of member checking has still not been fully resolved. In the response letter, the authors indicate that “weekly peer debriefings and member checking” were undertaken. However, the manuscript does not clearly explain what the member checking entailed, whether it involved participants directly, and at what stage it occurred. Please revise the manuscript to state explicitly whether member checking was conducted and, if so, briefly describe the process. If the study did not include participant/member checking in the conventional sense, please remove that terminology and instead describe the specific credibility procedures that were used.

Second, the Data Availability statement remains inconsistent across the submission materials. The response letter suggests a clearer controlled-access pathway for anonymized excerpted data, including institutional oversight and access procedures. However, the manuscript/front matter still appears to include older wording indicating that data are not publicly shared and referring only to the interview guide and coding framework in the supporting materials. Please ensure that the Data Availability statement is fully harmonized across the manuscript and submission metadata and clearly specifies what data are available, through what mechanism, under what conditions, and with what process for review and release.

Finally, please conduct one last careful consistency check of the clean manuscript before resubmission. The compiled editorial PDF suggests some remaining inconsistencies across versions, and all text, metadata, and supporting statements should now be fully aligned.

Reviewers' comments:

 Figure Resubmissions:

---

## [Editor Report · Decision Letter 3]

15 Apr 2026

PGPH-D-25-03150R3

Climate change, livelihoods, gender and violence in Rukiga, Uganda: intersections and pathways

Dear Dr. Muhumuza,

Thank you for submitting your manuscript to PLOS Global Public Health. After careful consideration, we feel that it has merit but does not fully meet PLOS Global Public Health’s publication criteria as it currently stands. Therefore, we invite you to submit a revised version of the manuscript that addresses the points raised during the review process.

We look forward to receiving your revised manuscript.

Kind regards,

Zahra Zeinali, MD MPH DrPH

Academic Editor

**Journal Requirements:**

**Additional Editor Comments (if provided):**

There is still a potentially important sample-size inconsistency. The manuscript says there were 28 FGDs with six to eight participants each, plus 40 KIIs. That would imply at least 168 FGD participants plus 40 KII participants. But the Results section says: “A total of 120 participants were included in the study.”

Reviewers' comments:

 **Figure Resubmissions:**

---

## [Editor Report · Decision Letter 4]

1 May 2026

Climate change, livelihoods, gender and violence in Rukiga, Uganda: intersections and pathways

PGPH-D-25-03150R4

Dear Dr. Muhumuza,

We are pleased to inform you that your manuscript 'Climate change, livelihoods, gender and violence in Rukiga, Uganda: intersections and pathways' has been provisionally accepted for publication in PLOS Global Public Health.

Best regards,

Zahra Zeinali, MD MPH DrPH

Academic Editor